**Editor:** Hamid Sharifi, HIV/STI Surveillance Research Center and WHO Collaborating Center for HIV Surveillance, Institute for Future Studies in Health, Kerman University of Medical Sciences, ISLAMIC REPUBLIC OF IRAN

# Intentional abortion and its associated factors among female sex workers in Iran: Results from national bio-behavioral surveillance-2020

**Ghobad Moradi**[1]☯, **Mohammad Mehdi Gouya**[2]☯, **Elnaz Ezzati Amini**[1]☯, **Sahar Sotoodeh Ghorbani**[3]‡, **Samaneh Akbarpour**[4]‡, **Bushra Zareie**[5]‡, **Neda Izadi**[3]‡, **Farzaneh Kashefi**[6]‡, **Yousef Moradi**[1]☯*

1 Social Determinants of Health Research Center, Research Institute for Health Development, Kurdistan University of Medical Sciences, Sanandaj, Iran, 2 Iranian Center for Communicable Diseases Control, Ministry of Health and Medical Education, Tehran, Iran, 3 Department of Epidemiology School of Public health and Safety, Shahid Beheshti University of Medical Sciences, Tehran, Iran, 4 Occupational Sleep Research Center, Baharloo Hospital, Tehran University of Medical Sciences, Tehran, Islamic Republic of Iran, 5 Department of Epidemiology, School of Public Health, Hamadan University of Medical Sciences, Hamadan, Iran, 6 Reproductive health, Population, Family and School Health Department, Ministry of Health and Medical Education, Tehran, Iran

☯ These authors contributed equally to this work.
‡ SSG, SA, BZ, NI and FK also contributed equally to this work.
* yousefmoradi211@yahoo.com

## Abstract

### Purpose

In addition to estimating the prevalence of intentional abortion in Iranian female sex workers (FSWs), this study identified related factors using the data of a national study.

### Methods

This cross-sectional study utilizes the third round of integrated bio-behavioral surveillance-III data in Iranian FSWs in December 2019 and August 2020, and 1515 Participants were selected in 8 geographically diverse cities in Iran. Logistic regression was performed using unweighted analysis to identify factors associated with intentional abortion. Stata software (version 14), respondent-driven sampling analyses, and R (version 4.1.2) was used for data analysis.

### Results

From 1390 participants with valid responses to the abortion question, 598 (37.3%; 95% CI: 32.43, 42.11%) reported intentional abortion during their life. According to the age groups, the highest prevalence of abortion was in the age group of 31 to 40 years (42.60%). In the multivariate logistic regression model, the marital status (divorced women (AOR = 2.05, 95% CI: 1.29, 3.27), concubines (AOR = 1.78, 95% CI: 1.02, 3.11)), work experience in brothels (AOR = 1.39, 95% CI: 1.04, 1.84), the type of sex (AOR = 2.75, 95% CI: 1.35,

**Data Availability Statement:** The data supporting this study's findings cannot be shared publicly because of ethical restrictions involving sensitive patient information. Data are available upon request from the Social Determinants of Health Research Center, Research Institute for Health Development, Kurdistan University of Medical Sciences after appropriate protocol submission to the institution's office of Human Research Ethics Committee (contact via sdhkurdistan@gmail.com) for researchers who meet the criteria for access to confidential data. Dr. Bakhtiar Piroozi is head of the center and is responsible for considering data requests. The Social Determinant of Health Research Center began its work in 2012. During two years this center published over 100 articles in international and national journals, and it has carried out many research projects in the public health issues. Often, these projects were funded by WHO, UNAIDS, UNDP, and Iranian Ministry of Health. This center has a Research Council Members for deciding about projects and related their data. This Research Council Members included several experts and researchers related to public health issues (https://muk.ac.ir/Page?pageId=8714).

**Funding:** This study was supported by Kurdistan University of Medical Sciences, Sanandaj, Iran, with the grant number R.MUK.REC.1398.132. The funding agency did not play any role in the planning, conducting, reporting, or deciding to submit the paper for publication.

**Competing interests:** The authors have declared that no competing interests exist.

**Abbreviations:** FSW, Female Sex Workers; IBBS-III, Integrated Bio-Behavioral Surveillance-III; RDS, Respondent-driven Sampling; SRH, Sexual and Reproductive Health; STI, Sexually Transmitted Infections.

5.58), the history of sexual violence (AOR = 1.54, 95% CI: 1.19, 2.01), and alcohol consumption (AOR = 1.53, 95% CI: 1.18, 2.01) were significantly associated with a history of intentional abortion.

## Conclusion

Intentional abortion's prevalence among Iranian FSWs has been much higher than that of the general female population in Iran, which is an alarming issue in the public health of this group and needs more effective interventions. In addition, alcohol consumption, working in a brothel, and being divorced are essential factors in increasing abortions among sex workers.

## Introduction

Sex work is the provision of sexual services for money or the equivalent. Sex workers may be male, female, or transgender. The boundaries of sex work are blurred from erotic shows without physical contact with the client to unprotected sex with large numbers of clients [1]. Female sex workers (FSWs) are stigmatized and marginalized around the world. They are generally not accepted and addressed as criminals, sexual perverts, and disease carriers [2]. Women sex workers often work in secret and away from society. The reason for this is the excessive control of these people by culture, family, and others because these people have always been considered a distinct part of society. In many organizations, these people are primarily seen as a threat to these values due to different religious laws and cultures. These factors cause a lack or difficulty in accessing health, care, and treatment services and ultimately reduce the desire of these people to receive these services [3]. In addition, these factors have made FSWs a vital health challenge for health policymakers and public health professionals.

Providing services to this community is very important to prevent various diseases and their transmission. To provide services to this community, it is crucial to estimate the number of these people in different countries. According to the results of previous studies, the number of FSWs in various regions worldwide has a different range. The prevalence of FSWs (with 15–49 age) varies from 0.2% to 2.6% in Asia, 0.4% to 4.3% in sub-Saharan Africa, and 0.2% to 7.4% in Latin America [1, 4–6]. According to previous studies, approximately 40 to 42 million sex workers women worldwide; about 80% are women between 13 and 25 [7, 8]. The high amount of social, environmental, and fundamental problems such as HIV and other sexually transmitted infections (STIs), physical and sexual violence, substance and alcohol abuse, inappropriate use or non-use of condoms during sexual intercourse, multiple sexual partners at the same time and limited access to health care services affect the development and outcome of pregnancy in the FSWs population and increase undesirable sexual and reproductive health (SRH) and adverse pregnancy outcomes in this population [9–12].

Various studies have shown that unwanted pregnancies are prevalent among FSWs in different settings and often lead to intentional abortion [12–14]. The illegality of intentional abortion has led to many abortions being performed unsafe, which usually has many side effects. Studies in Iran also indicate about eight abortions annually per 1000 women of reproductive age [15–19]. However, induction abortion is not uncommon. For example, in 2012, the annual induced abortion rate was estimated to be approximately 8.9 per 1000 women in the general population [20, 21]. A 2010 study of FSWs in Iran estimated the annual abortion rate at 20.7 per 1000 women. However, the illegality of abortion and sex work has caused more problems

in preventing intentional abortion [20, 21]. In low- and middle-income countries, abortion is a religiously illegal or immoral act [22]. Therefore, access to drugs required for abortion, such as misoprostol or other pre- or post-abortion care services, is difficult for all sex workers in these countries. But the class gap in these countries in terms of socioeconomic status can make access to abortion-related health and care services different for FSWs. For example, FSWs living in a higher class or with a better socioeconomic status have easier access to the drugs needed to perform an abortion and even post-abortion care (PAC) services [23–25]. While FSWs are in the lower socioeconomic classes, they are more likely to have high-risk abortions and deaths.

On the other hand, in low- or middle-income societies, age is another important and influential factor in causing unsafe abortion and its complications. According to previous studies, young FSWs are more likely to have unsafe abortions. Young women and adolescents between 15 and 24 carry a significant burden of low-risk abortions in developing areas, accounting for 41% of unsafe abortions. In general, information about abortion in sexually active women is incomplete and limited globally. Still, the results show that sub-Saharan Africa has a high rate of abortion in these groups. In Côte d'Ivoire, more than half of FSWs report not continuing their pregnancies to the end. Also, 47.7% of FSWs in Zambia have reported this [25, 26]. Like in other developing countries, FSWs in Iran are subject to social restrictions.

According to the new laws in Iran, abortion is considered illegal, except in cases where the mother's life is in danger during pregnancy. Therefore, for these societies, abortion is practically not legally possible, and for this reason, illegal and dangerous abortions are performed on Iranian FSWs. According to the results of studies, illegal abortion in Iranian women of reproductive age is 8 cases per 1000 people. Necessary measures to perform legal abortion and study the conditions of abortion in women of reproductive age in Iran in the socio-economic conditions can be done very little. Still, for at-risk groups such as FSWs who are primarily stigmatized and discriminated against socially because of their activities, these legal measures are practically impossible [25–28]. On the other hand, high-risk sexual behaviors such as not using condoms, sexual violence during sexual intercourse, sexual intercourse in cities far from home, lack of access to appropriate contraceptive methods, and anal sex are other factors. These factors are also present in Iran's neighboring countries. For example, in a study conducted in Afghanistan, the results showed that anal sex and violent sex are among the factors influencing the occurrence of miscarriage in FSWs [25, 26].

To reduce harm and design effective interventions in this group, it is valuable and necessary to provide educational services to improve the reproductive health of FSWs and identify the factors affecting and preventing abortion. Therefore, this study aimed to determine the prevalence of intentional abortion and its characteristics in Iranian FSWs.

## Methods

### Study design, setting, and participants

In this cross-sectional study, the third round of bio-behavioral surveillance data in the group of FSWs, called integrated bio-behavioral surveillance-III (IBBS-III), was performed in December 2019 and August 2020. The IBBS-III survey was conducted by the respondent-driven sampling (RDS) method in eight major cities of Iran, including Sari, Tabriz, Tehran, Bandar Abbas, Shiraz, Mashhad, Kermanshah, and Khorramabad, with the possibility of creating the highest variation in sampling. The RDS sampling is the best method to better and proper access to the hidden population in epidemiology studies [29]. To conduct the study by the RDS method, first, the study was started according to the sample size specified for each city and by selecting a small number of initial participants known as "seeds"; then, they were asked

to use referral coupons to enter in a chain manner other people from their peers who had the conditions to join the study. Inclusion criteria also included age 16 years or older, having penetrative sex (vaginal/anal) for money with more than one male client in the past year, living or working in the selected city for at least one to two months after the interview, having a valid coupon (RDS except for headers) and giving initial consent to participate in the study. Finally, 1515 FSWs participated in the study (more details are in the press in the article "Behavioral and serological survey of HIV/AIDS prevalence among female sex workers in Iran: A national study using respondent-driven sampling-2020").

## Data collection

First, written informed consent was obtained from individuals to participate in the study, and then face-to-face interviews were conducted in a private room by a trained female interviewer. To collect the data, the general questionnaire of the behavioral and serological survey on FSWs, including information on intentional abortion as well as demographic, social, and behavioral characteristics such as the history of sexual intercourse with clients, condom use, the experience of sexual violence, and alcohol and drug use was applied. After an almost one-hour interview, participants received monetary incentives.

## Dependent variable

The dependent variable was the answer to the question "Have you ever had an intended abortion," which was a binary variable (yes/no) and considered as the main outcome of the study.

## Covariates

Independent variables included the age, nationality (Iranian, or non-Iranian), marital status (single, married, divorced, concubine, widow, and partner), level of education (illiterate, elementary, middle school, high school, diploma, and university), age in the first vaginal or anal sex, age in the first sex as a prostitute, attending team houses and hangouts to have sex with clients or finding clients (yes, no), group sex experience (yes, no), number of clients in the past month (1, 2–5 and equal / more than 6 clients), type of sexual contacts with clients (vaginal and anal / oral), rate of condom use in sexual relations with clients in the past month (all times, most of the time, sometimes and never), use of condoms in the last vaginal, anal or oral sex contact (yes, no), the person proposing the use of condoms in sexual contacts (myself, the client and by mutual agreement), level of education of clients (Illiterate, undergraduate, diploma, university and I do not know), main way of client acquisition (team houses / hangouts, referrers (owners and pimps), cyberspace (via mobile, internet and social networks) and others (parties, shopping centers, streets, parks, introduction through friends, hotels, inns and public transportation), experience of sexual violence (yes, no), test result of HIV serology (positive, negative), experience of lifetime alcohol consumption (yes, no), experience of lifetime drug use (yes, no) and experience of injecting drugs so far (yes, no).

## Data entry and analysis

In the present study, the data were first examined for missing data and the possibility of errors in entering them into the software, and after clearing them, the data for all cities were merged. Finally, 1390 people who answered the question related to lifetime intentional abortion were included in the analysis.

Also, according to the selection of samples in this study, i.e., using the RDS method, weighted analysis was used in RDS Analyst software in such a way that all variables were

weighted considering the weight equivalent to the network size for each person (the number obtained from the question "How many women aged 16 years or older do you know, who have had sex with a male customer at least once in the last 12 months in exchange for money, drugs or anything else?" The previous texts limited minimum and maximum network sizes to 3 and 150 people [30]. According to Giles's method, weighting was performed, and the initial population was estimated at 90,000 samples for weighting.

Frequencies in the tables were obtained based on the sample. Weighted percentages in the contingency tables for the target population were calculated based on the RDS-II method and Bootstraps with the number 1000. Also, P-value in the contingency tables was assigned to the weighted frequencies calculated by RDS software. We did not report non-weight percentages for three reasons. First, these percentages are less valid than weight percentages. Second, the tables should not be confusing to readers, and third, non-weight percentages can be calculated through the reported frequencies. The P-value obtained for qualitative and quantitative variables was based on Pearson's Chi-squared and t-test, respectively. Variables significantly associated with intentional abortion in cross-tabulations at the level of P-Value = 0.05 were selected for logistic regression analysis. In selecting variables for multivariate analysis, we had no restrictions except for variables that have collinearity with each other base on correlation coefficient. We selected the variables that had the highest correlation with the outcome for multivariate analysis from among the variables that had a correlation with each other.

Univariate and multivariate logistic regression was performed using non-weighted analysis based on the suggestion in the latest article [31]. Stata software (version 14), RDS Analyze, and R (version 4.1.2) were used to perform all analyzes.

## Results

The prevalence of lifetime intentional abortion based on the demographic characteristics of FSWs was presented in Table 1. The prevalence of lifetime intentional abortion in 1390 women was 37.3%; 95% CI: 32.43, 42.11%. According to the age group, the prevalence of abortion was higher in 31 to 40 years (42.60%) and then in less than 30 years (34.16%). Based on marital status, the highest prevalence of abortion was reported among divorced women (42.92%). Regarding education, the highest prevalence of abortion was reported in women with the secondary education level (63.20) and then in ones with diplomas (60.26). FSWs who reported lifetime intentional abortion were younger than those who did not have an abortion in their first sexual intercourse (P-value = 0.014) (Table 1).

Table 2 shows the prevalence of lifetime intentional abortion by the type of high-risk behaviors. The frequency of lifetime intentional abortion among FSWs who worked in brothels was 44.55%, significantly higher than its frequency among FSWs who did not work in brothels (P-value <0.001). Also, the frequency of lifetime intentional abortion in FSWs who had group sex and vaginal sex was 43.62% and 38.49%, respectively, significantly higher than in FSWs who did not have such high-risk behaviors.

FSWs who have experienced sexual violence had significantly more abortions than other populations of FSWs (45.45% vs. 33.5%; P-value <0.001). FSWs with a history of alcohol consumption (42.92%), substance abuse (44.22%), and injecting drug use (59.8%) reported a history of intentional abortion significantly higher than FSWs who did not have such high-risk behaviors (Table 2).

Table 3 shows the information on the study of factors associated with life time intentional abortion in the studied FSWs using logistic regression. In the multivariate logistic regression model, the marital status (divorced women (AOR = 2.05, 95% CI: 1.29, 3.27), and concubines (AOR = 1.78, 95% CI: 1.02, 3.11)), work experience in brothels (AOR = 1.39, 95% CI: 1.04,

**Table 1. Demographic characteristics and intentional abortion among female sex workers in Iran.**

| Variables | N = 1390 | Status of intentional abortion, n (%) | | P-value |
|---|---|---|---|---|
| | | No N = 792 62.7 (57.89, 67.57 | Yes N = 598 37.3 (32.43, 42.11) | |
| **Age mean ± SD** | 35.78 (9.23) | 35.82 (9.73) | 35.71 (8.35) | 0.825 |
| **Age** | | | | |
| **≤30 years** | 443 (31.94) | 277 (65.84) | 166 (34.16) | **<0.001** |
| **31–40 years** | 556 (39.66) | 281 (57.40) | 275 (42.60) | |
| **41–50 years** | 322 (23.17) | 189 (64.88) | 133 (35.12) | |
| **51≥ years** | 69 (5.23) | 45 (74.61) | 24 (25.39) | |
| **Nationality** | | | | |
| **Iranian** | 1373 (98.41) | 784 (62.90) | 589 (37.10) | 0.409 |
| **Non-Iranian** | 16 (1.59) | 7 (51.97) | 9 (48.03) | |
| **Marital status** | | | | |
| **Single** | 135 (10.01) | 94 (79.48) | 41 (20.52) | **<0.001** |
| **Married** | 302 (25.22) | 185 (64.17) | 117 (35.83) | |
| **Divorced** | 630 (46.12) | 331 (57.08) | 299 (42.92) | |
| **Concubine** | 164 (8.26) | 83 (61.69) | 81 (38.31) | |
| **Widow** | 110 (8.20) | 69 (68.63) | 41 (31.37) | |
| **Living with partner** | 41 (2.18) | 24 (71.61) | 17 (28.39) | |
| **Education level** | | | | |
| **Illiterate** | 102 (8.32) | 66 (74.87) | 36 (25.13) | **0.050** |
| **Elementary** | 290 (22.39) | 156 (59.09) | 134 (40.91) | |
| **Middle school** | 346 (25.95) | 201 (63.20) | 145 (36.80) | |
| **High school** | 162 (10.68) | 95 (62.38) | 67 (37.62) | |
| **Diploma** | 350 (23.64) | 187 (60.26) | 163 (39.74) | |
| **Academic** | 139 (9.01) | 87 (67.31) | 52 (32.69) | |
| **Age at first sexual contact (years)** | 17.31 (4.22) | 17.52 (4.48) | 16.96 (3.74) | **0.014** |

1.84), the type of sex (AOR = 2.75, 95% CI: 1.35, 5.58), the history of sexual violence (AOR = 1.54, 95% CI: 1.19, 2.01), and alcohol consumption (AOR = 1.53, 95% CI: 1.18, 2.01) had a significant association with the history of life time intentional abortion in this population (Table 3).

## Discussion

The present study estimated the prevalence of intentional abortion in FSWs in Iran and evaluated its related risk factors. The prevalence of lifetime intentional abortion among FSWs was 37.3%; 95% CI: 32.43, 42.11%. The proportion of intentional abortion was higher in divorced women and concubines. Also, abortion was more elevated in FSWs who worked in brothels and had a history of sexual violence and alcohol consumption. These women are more likely than other sex workers to become pregnant and eventually have an abortion due to increased clients and multiple sexual partners.

On the other hand, FSWs who work in hangouts or brothels face the dilemma of choosing between using contraception or reducing emotional and high-income sex. These women must resist stigma and social discrimination if they choose to use contraception. If they decide to decline their sexual partners and clients, they must face reduced demand and income. Therefore, they often have to select the second option, so they are more prone to pregnancy and abortion [32–34]. These conditions are more in divorced women than in single women. Also, FSWs in brothels are more likely than other FSWs to use drugs such as amphetamines or glass

**Table 2. Sexual behavior and intentional abortion among female sex workers in Ir.**

| Variables | N = 1390 | Status of intentional abortion, n (%) | | P-Value |
|---|---|---|---|---|
| | | No | Yes | |
| **Age of first prostitution mean ±SD** | 27.59 (8.56) | 27.62 (8.66) | 27.54 (8.39) | 0.867 |
| **History of worked in team houses/hangouts** | | | | |
| Yes | 612 (31.35) | 298 (55.45) | 314 (44.55) | **<0.001** |
| No | 771 (68.65) | 494 (66.13) | 277 (33.87) | |
| **Group sex** | | | | |
| Yes | 376 (16.17) | 178 (56.38) | 198 (43.62) | **0.038** |
| No | 1002 (83.83) | 610 (64.02) | 392 (35.98) | |
| **Number of clients (past month)** | | | | |
| 1 | 73 (6.68) | 44 (56.05) | 29 (43.95) | 0.088 |
| 2–5 | 456 (38.61) | 268 (66.13) | 188 (33.87) | |
| ≥6 | 707 (48.82) | 393 (61.69) | 314 (38.31) | |
| Do not know | 106 (5.89) | 56 (54.31) | 50 (45.69) | |
| **Types of sexual contact** | | | | |
| Vaginal | 1276 (95.45) | 708 (61.51) | 568 (38.49) | **<0.001** |
| Anal/ Oral | 62 (4.55) | 50 (85.16) | 12 (14.84) | |
| **Frequent use of condoms (past month)** | | | | |
| All the time | 545 (49.48) | 311 (62.96) | 234 (37.04) | **0.066** |
| most of the time | 223 (13.56) | 118 (59.39) | 105 (40.61) | |
| Sometimes | 387 (21.90) | 207 (58.41) | 180 (41.59) | |
| Never | 186 (15.06) | 124 (69.90) | 62 (30.10) | |
| **Using of condom in last sex (Vaginal, anal or oral)** | | | | |
| Yes | 944 (69.94) | 542 (63.11) | 402 (36.89) | 0.960 |
| No | 438 (30.06) | 247 (62.81) | 191 (37.19) | |
| **Who suggested using of condom** | | | | |
| Myself | 692 (84.73) | 391 (62.13) | 301 (37.87) | 0.189 |
| Customer | 42 (3.13) | 26 (67.47) | 16 (32.53) | |
| By agreement of each other | 99 (12.14) | 65 (73.62) | 34 (26.38) | |
| **Education level of the sexual partner** | | | | |
| Illiterate | 90 (5.76) | 43 (60.14) | 47 (39.86) | |
| Less than a high school | 268 (18.27) | 166 (69.08) | 102 (30.92) | **0.027** |
| Diploma | 387 (22.21) | 222 (64.96) | 165 (35.04) | |
| Academic | 211 (13.83) | 127 (65.66) | 84 (34.34) | |
| Do not know | 428 (39.93) | 231 (58.10) | 197 (41.90) | |
| **The main way to find clients currently** | | | | |
| Hangout | 119 (5.34) | 53 (58.46) | 66 (41.54) | **<0.001** |
| Pimp | 321 (21.93) | 161 (51.60) | 160 (48.40) | |
| Cyberspace (phone, internet) | 252 (18.47) | 151 (63.80) | 101 (36.20) | |
| Others (party, shopping center, streets, friends, hotel, etc.) | 689 (54.26) | 425 (67.99) | 264 (32.01) | |
| **Experience of sexual violence** | | | | |
| Yes | 558 (30.42) | 265 (54.55) | 293 (45.45) | **<0.001** |
| No | 826 (69.58) | 524 (66.52) | 302 (33.48) | |
| **HIV serostatus** | | | | |
| Negative | 1369 (98.47) | 777 (62.41) | 592 (37.59) | 0.077 |
| Positive | 21 (1.53) | 15 (83.52) | 6 (16.48) | |
| **Ever consumed alcohol** | | | | |
| Yes | 820 (52.32) | 425 (57.08) | 395 (42.92) | **<0.001** |

(*Continued*)

**Table 2.** (Continued)

| Variables | N = 1390 | Status of intentional abortion, n (%) | | P-Value |
|---|---|---|---|---|
| | | No | Yes | |
| No | 522 (47.68) | 336 (67.62) | 186 (32.38) | |
| **Ever used drugs** | | | | |
| Yes | 476 (29.25) | 242 (55.78) | 234 (44.22) | <**0.001** |
| No | 881 (70.75) | 533 (65.67) | 348 (34.33) | |
| **Ever injected drugs** | | | | |
| Yes | 43 (2.06) | 17 (40.22) | 26 (59.78) | **0.024** |
| No | 1314 (97.94) | 758 (63.25) | 556 (36.75) | |

and alcohol during sex. This leads to unprotected sex and violence in these groups, the important consequences of which can be pregnancy and eventually abortion [32, 35, 36]. This result indicated the need for special attention from health policymakers and health professionals in this area.

In a previous study, the prevalence of intentional abortion among Iranian FSWs was 35.3%, so intentional abortion among FSWs in Iran has increased by approximately 2% between 2010 and 2020 [20, 21]. In the study of Khezri et al. conducted in 13 provinces of Iran, the prevalence of abortion in Iranian FSWs was 46.5%, while the prevalence in the present study was 37.3%. One of the essential reasons for this difference in the results of these two studies, both of which have been conducted in Iran at different times, is the difference in the selected provinces and how to report or calculate the prevalence of abortion [27]. In the study by Khezri et al., 13 provinces were selected, while in the present study, eight provinces were surveyed. The provinces that may have the highest prevalence of female FSWs were studied in a study by Khezri et al. On the other hand, the prevalence of weight was used to report abortion in the present study. In contrast, the study of Khezri et al. used the raw prevalence to report abortion [27].

The prevalence of intentional abortion among FSWs in Iran is comparable to those in different international regions [30, 31, 37, 38]. In addition, this prevalence varied from 21.4% to 40.0% in Cambodia [39, 40], up to 51% in China (18), and 53% in Colombia [41]. Also, in Hong Kong, 58% of FSWs reported a history of intentional abortion [42]. Therefore, the result of the present study in Iran is in line with those of studies conducted abroad.

Concerning internal studies, the research conducted by Erfani et al. on the general population showed that from 2010 to 2015, the rate of intentional abortion decreased from 5.5 to 4.4 per 1,000 women. The annual number of abortions also reduced from 11,500 to 11,400 [43].

Of course, it should be noted that the prevalence of intentional abortion in general populations varies from country to country. For example, in a 2010 study in the Nigerian metropolitan area, Okonofua estimated the overall abortion rate at 49.6%, of which about 82.2% was intentional abortion [44], or in another study by Bernabé-Ortiz et al. (2009) on women in Latin America, they estimated the prevalence of intentional abortion at 11.6% [45]. Although the prevalence of intentional abortion in different populations of the world showed different prevalence and a wide range, it should be noted that the results of studies in Iran have shown that the prevalence of intentional abortion was about 5.5% in the general female population [45, 46]. However, in the present study, the prevalence of intentional abortion in FSWs was approximately 37.3%, much higher than that in the general population. The prevalence of intentional abortion in Iranian FSWs is higher than that in women of the general population, which indicates the need for significant attention from policymakers [15]. The prevalence of abortion in this Iranian community indicates the high incidence of unwanted and

**Table 3. Factors associated with intentional abortion among female sex workers in Iran using logistic regression (n = 1234).**

| Variables | Crude OR (95% CI) | P-value | Adjusted OR | P-value |
|---|---|---|---|---|
| Age (yr.) † | 1.01 (0.99, 1.02) | 0.329 | 1.01 (0.99, 1.02) | 0.239 |
| Education level †† | 0.99 (0.92, 1.07) | 0.838 | 0.97 (0.88, 1.06) | 0.489 |
| **Marital status** | | | | |
| **Single** | Ref. | | Ref. | |
| **Married** | 1.45 (0.94, 2.24) | 0.093 | 1.62 (0.98, 2.70) | 0.062 |
| **Divorced** | 2.07 (1.39, 3.09) | <0.001 | 2.05 (1.29, 3.27) | **0.003** |
| **Concubine** | 2.24 (1.39, 3.61) | 0.001 | 1.78 (1.02, 3.11) | **0.042** |
| **Widow** | 1.36 (0.80, 2.32) | 0.255 | 1.38 (0.72, 2.64) | 0.336 |
| **Living with partner** | 1.62 (0.79, 3.34) | 0.188 | 1.22 (0.56, 2.68) | 0.621 |
| Age at first sexual contact (years) † | 0.97 (0.94, 0.99) | 0.05 | - | - |
| **History of worked in team houses/hangouts** | | | | |
| **No** | Ref. | | Ref. | |
| **Yes** | 1.88 (1.51, 2.33) | <0.001 | 1.39 (1.04, 1.84) | **0.024** |
| **Group sex** | | | | |
| **No** | Ref. | | Ref. | |
| **Yes** | 1.73 (1.36, 2.20) | <0.001 | 1.25 (0.93, 1.67) | 0.142 |
| **Types of sexual contact** | | | | |
| **Anal/ Oral** | Ref. | | Ref. | |
| **Vaginal** | 3.34 (1.76, 6.34) | <0.001 | 2.75 (1.35, 5.58) | **0.005** |
| **experience of sexual violence** | | | | |
| **No** | Ref. | | Ref. | |
| **Yes** | 1.92 (1.54, 2.39) | <0.001 | 1.54 (1.19, 2.01) | **0.001** |
| **Ever consumed alcohol** | | | | |
| **No** | Ref. | | Ref. | |
| **Yes** | 1.68 (1.34, 2.10) | <0.001 | 1.53 (1.18, 2.01) | **0.002** |
| Ever used drugs* | | | | |
| **No** | Ref. | | - | - |
| **Yes** | 1.48 (1.18, 1.85) | 0.001 | - | - |
| Ever injected drugs* | | | | |
| **No** | Ref. | | Ref. | |
| **Yes** | 2.9 (1.12, 3.88) | 0.020 | 1.63 (0.85, 3.15) | 0.143 |
| **Education level of the sexual partner** | | | | |
| **Academic** | Ref. | | Ref. | |
| **Diploma** | 1.12 (0.80, 1.58) | 0.503 | 0.89 (0.60, 1.32) | 0.556 |
| **Less than a high school** | 0.92 (0.64, 1.34) | 0.696 | 0.89 (0.57, 1.39) | 0.615 |
| **Illiterate** | 1.65 (1.01, 2.72) | 0.048 | 1.21 (0.65, 2.25) | 0.553 |
| **Don't know** | 1.29 (0.92, 1.80) | 0.137 | 1.32 (0.88, 1.97) | 0.178 |
| **The main way to find clients Currently** | | | | |
| **Cyberspace (phone, internet)** | Ref. | | Ref. | |
| **Hangout** | 1.86 (1.20, 2.89) | 0.006 | 1.25 (0.75, 2.07) | 0.399 |
| **Pimp** | 1.49 (1.06, 2.07) | 0.020 | 1.17 (0.80, 1.70) | 0.415 |
| **Others (party, shopping center, streets, friends, hotel, etc.)** | 0.93 (0.69, 1.25) | 0.623 | 1.00 (0.72, 1.37) | 0.985 |

† Because there is collinearity between *age* and *age at first sexual contact*, only one variable was selected for the adjusted model based on the maximum of crud OR.

* Because there is collinearity between *used drugs* and *injection drugs*, only one variable was selected based on the maximum crud OR for the adjusted model.

††The level of *education* was coded as the highest level (basic level) to the lowest level

OR: Odds Ratio, CI: Confidence Interval

unintentional pregnancies in this key population, which requires particular intervention as soon as possible to provide the necessary training for contraception, and how to use contraceptives; finally, to prevent unwanted pregnancies. The findings of the present study and other studies in Iran on this crucial population to determine the prevalence of abortion can support the hypothesis that the population of FSWs needs the provision of training services and free contraceptives. So, these results can show the importance of using condoms and other contraceptives in this group. For example, this study showed that most FSWs who had an intentional abortion did not use a condom, although this difference was not statistically significant. The use of condoms was also examined in this study. The results have shown that 41.59% of FSWs sometimes use condoms, and this can be considered an alarm for health policymakers because it indicates a decrease in using condoms in this key population, which can also increase the transmission of sexually transmitted diseases in addition to improving the adverse outcomes of pregnancy in this group.

The present study results showed that divorced or widows had a higher probability of intentional abortion than married FSWs. Divorced women cannot have children in Iran and neighboring countries due to different and strict cultures. On the other hand, if this happens, they will be severely stigmatized and discriminated against by people around them and society, which increases the probability of intentional abortion in divorced or widows FSWs [47]. Also, according to the results, concubines are more likely to have an abortion than single women; this finding is consistent with the findings of other studies in this field, showing that FSWs with a fixed partner are more likely to have a miscarriage [39, 48]. There are several hypotheses about the relationship between abortion and the type of sex. Do Iranian FSWs tend to have other types of sex to avoid pregnancy? Is there a history of different kinds of sex for more pleasure, a tendency to experience a variety of behaviors in riskier FSWs, and not paying attention to contraception? The results confirm the first hypothesis. The odds ratio of having an abortion in a vaginal contact is 2.75 times that of FSWs who experience other contacts more than vaginal contact. Working in a brothel leads to less access to contraceptive methods, exposure to many clients, and ultimately multiple sexual partners. This increases the likelihood of high-risk and violent sex in FSWs, leading to improved pregnancy and abortion [48, 49].

In the study of the association between the studied variables and intentional abortion, it was found that alcohol consumption had a significant association with intentional abortion in the population of FSWs, so lifetime alcohol consumption increased intentional abortion in this population. Consumption of alcohol also has special conditions because it is banned in Iran. We examined the relationship between alcohol and abortion in response to whether people who have a history of alcohol use have specific characteristics compared to people who do not drink alcohol that predispose them to pregnancy and abortion? The results of our study were consistent with the ones of a 2010 study by Katherine et al. They also found in their study examining the factors associated with the termination of pregnancy among FSWs in Afghanistan that the probability of intentional abortion increased with lifetime alcohol consumption or illicit drug use and having one unforeseen pregnancy [50]. As reported in other studies, alcohol consumption may interfere with the proper and consistent contraceptives, especially condoms [50–52]. In high-risk groups such as the FSWs, most unsafe abortions can be attributed to a reluctance to care for the child, the negative impact of having a child on the mother's relationships and occupation, economic and social problems, stigma, discrimination due to unborn father, and attributed to the fear of declining incomes. In developing countries, especially Iran, there is a need to integrate the family planning program with all sectors of education and care programs in the national health plans of the country. Because the best way to prevent unwanted pregnancies and their related consequences is to provide and distribute modern contraceptive services and provide the necessary and appropriate care for pregnant

FSWs. FSWs are groups that face stigma and discrimination in society, and the fear of this discrimination prevents them from going to centers and receiving modern methods of contraception. The right solution is to set up family planning centers and contraceptive methods for this community, especially the younger FSWs. This is because younger FSWs are at greater risk for unwanted pregnancies and unsafe abortions due to la ack of experience and having a wide variety of clients [23, 53–56].

Finally, it can be imagined that the environments in which FSWs do their works and activities reflect the story and living conditions of these populations from childhood until now, which reflects the social unrest and marginalization [56–58]. These people are exposed to sexual abuse, work, stigma, discrimination, and social violence that can directly impact mental disorders and substance and alcohol abuse in these groups [54, 59]. Given the challenges these people face, it can be said that these women's flexibility and tolerance threshold are very high because they always hope to have a better life. Therefore, performing health, care, and treatment interventions in these groups may significantly impact. For example, a study conducted in Kenya showed that FSWs could communicate and interact appropriately, especially to convince their sexual partners to use a condom. Even these people have motivated themselves and their sexual partners to use the necessary antiviral and bacterial drugs before or after sexual intercourse [60]. Another study also shows that FSWs, to support themselves and their peers, have set up a fund to provide the necessary support. On the other hand, to reduce harm from society, they have increased their friendly relations with their fellow human beings. By forming different groups and establishing ties with humanitarian organizations, they have created a supportive atmosphere for themselves and their fellow human beings [53, 55].

Therefore, health policymakers can develop and implement training programs to increase the awareness of these groups by emphasizing the risk factors related to abortion, such as alcohol consumption, working in team houses, and being divorced. On the other hand, health policymakers can design and implement support programs for these groups, such as empowerment programs, holding support courses, providing social activities, and forming support organizations, to better adhere to the treatment of these people, programs Care and health, better participation of these people in educational programs to help.

The present study had some limitations. The most important limitation was related to the participants' self-report. Due to the illegality of abortion, women may have hidden abortion reports out of fear, so the results have led to underreporting. If this assumption is correct, the prevalence of intentional abortion is expected to be even higher than the number reported in the present study. The results of the present study, together with the results of previous studies in Iran, can provide a complete picture of the status of FSWs for health policymakers to develop care, support, and treatment plan.

## Conclusion

Based on this result, the prevalence of intentional abortion in FSWs is higher than in the general female population aged 15–50 years, which indicates a warning issue in the country's public health and needs the great attention of policymakers. In addition, alcohol consumption, working in team houses, and being divorced are essential factors in increasing abortions in sex workers.

Therefore, in providing services to this group, special attention should be paid to preventing unwanted pregnancies and reducing damages caused by the prevalence of intentional abortion. So, if the health policymakers decided to perform interventions in this population, the risk factors that increase the chance of intentional abortion in this study should be considered.

## Acknowledgments

We would like to thank all the Iranian Ministry of Health and Medical Education staff and individuals who helped us complete this research project.

## Author Contributions

**Conceptualization:** Yousef Moradi.

**Data curation:** Yousef Moradi.

**Formal analysis:** Sahar Sotoodeh Ghorbani, Samaneh Akbarpour, Bushra Zareie, Neda Izadi, Farzaneh Kashefi, Yousef Moradi.

**Funding acquisition:** Yousef Moradi.

**Investigation:** Sahar Sotoodeh Ghorbani, Samaneh Akbarpour, Bushra Zareie, Neda Izadi, Farzaneh Kashefi, Yousef Moradi.

**Methodology:** Sahar Sotoodeh Ghorbani, Samaneh Akbarpour, Bushra Zareie, Neda Izadi, Farzaneh Kashefi, Yousef Moradi.

**Project administration:** Farzaneh Kashefi, Yousef Moradi.

**Resources:** Sahar Sotoodeh Ghorbani, Samaneh Akbarpour, Bushra Zareie, Neda Izadi, Farzaneh Kashefi, Yousef Moradi.

**Software:** Sahar Sotoodeh Ghorbani, Samaneh Akbarpour, Bushra Zareie, Neda Izadi, Farzaneh Kashefi, Yousef Moradi.

**Supervision:** Ghobad Moradi.

**Validation:** Ghobad Moradi.

**Visualization:** Ghobad Moradi.

**Writing – original draft:** Ghobad Moradi, Mohammad Mehdi Gouya, Elnaz Ezzati Amini, Yousef Moradi.

**Writing – review & editing:** Ghobad Moradi, Mohammad Mehdi Gouya, Elnaz Ezzati Amini.

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
