## [Decision Letter · Decision Letter 0]

11 Apr 2022

PONE-D-22-07026Intentional abortion and its associated factors among female sex workers in Iran: results from national bio-behavioral surveillance-2020PLOS ONE

Dear Dr. Moradi

Thank you for submitting your manuscript to PLOS ONE. After careful consideration, we feel that it has merit but does not fully meet PLOS ONE’s publication criteria as it currently stands. Therefore, we invite you to submit a revised version of the manuscript that addresses the points raised during the review process.

We look forward to receiving your revised manuscript.

Kind regards,

Hamid Sharifi

Academic Editor

PLOS ONE

Journal Requirements:

Additional Editor Comments:

Dear Dr. Moradi,

Thanks so much for your submitting the manuscript to PLOS ONE. Based on the reviwers' comments and my assessment. You need to revise the work and resubmit it for further consideration.

Reviewers' comments:

Reviewer's Responses to Questions

**Comments to the Author**

1. Is the manuscript technically sound, and do the data support the conclusions?

Reviewer #1: Yes

Reviewer #2: Partly

2. Has the statistical analysis been performed appropriately and rigorously? 

Reviewer #1: I Don't Know

Reviewer #2: No

3. Have the authors made all data underlying the findings in their manuscript fully available?

Reviewer #1: Yes

Reviewer #2: No

4. Is the manuscript presented in an intelligible fashion and written in standard English?

Reviewer #1: No

Reviewer #2: Yes

5. Review Comments to the Author

Reviewer #1: I would like to acknowledge the tenacity and drive required to undertake this type of research. This manuscript tackles an important Public Health issue worldwide. However, the manuscript also presents major deficiencies, some of which I have detailed below:

The English usage in the manuscript should be thoroughly revised.

Abstract

Suggesting to point out the “Background” instead of “Purpose” and add 1 or 2 lines of the background of the study.

Page 2 line 30 “Participants” is started with the capital word

Page 2 line 38 Mention “type of sexual activity” instead of “type of sex”

Introduction:

The introduction part needs to be improved with proper linking of background.

The Social impact of abortion in FSWs in Iran should be explained.

The status of intentional abortion in FSWs in some neighboring countries should be mentioned.

Is there any current surveillance of FSWs available in Iran? If yes, please mention it with function.

Is there any current intervention or Government policy providing a better life to FSWs in Iran? Please briefly describe it.

Page 3 line 55 What type of services did you mention?

Page 3 line 61 add “age”

Page 4 line 76 mention the range of reproductive age

Methods:

What’s the cause of selecting the RDS method in sampling? What’s the difference of it among the other sampling procedures?

In which considerations you select 8 cities? What’s the proportion of the FSWs in these cities?

How do you check the collinearity? Explain it in the Data analysis part

Page 7 line 133 mention the software name

Page 7 line 149 P-Value “≤ 0.05”

Page 7 line 152 Please mention which software was used for which analysis separately.

Results:

Page 8 line 160 add “%” after values.

Page 11 line 181 Please elaborate the “AOR”

Page 12 line 183 Mention “type of sexual activity” instead of “type of sex”

Discussion:

Page 14 line 195 Replace the line “This result was indicated the …….. professionals in this area” after the “In a previous study…… 2 % between 2010 and 2020”

Page 14 line 198 What’re the possible causes to increase the prevalence in Iran? Mention it.

Conclusion:

Mention some significant risk factors of the current study.

References

Please check the reference style of the journal and maintain the format in all references.

Table

Mention the elaboration of the shorts forms below the tables that were used in tables

Reviewer #2: This topic is interesting due to limited attention to the sexual and reproductive health needs of female sex workers worldwide. However, I have several concerns to be addressed to refine the manuscript. My main concern is that this replication of the analysis of the previous surveys on the history of induced abortion among FSW is not a significant contribution to the literature and does not add much to what we already know. I’d consider adding to the depth of the analysis and also evaluating contraceptive use practices of FSW in Iran. Please see my comments in the following.

1. Overall, the paper is poorly written. I would suggest requesting a style and grammar review before submitting it to the journal.

Abstract

2. Authors stated that "In addition to estimating the prevalence of intentional abortion in Iranian Female sex 27 workers (FSWs), this study identified related factors using the data of a national study." As far as I know, two previous studies on female sex workers in Iran examined abortion and its associated factors. What is the added value of this paper to our understanding of abortion in this population in Iran?

3. "….in December 2019 and August 2020….". It should be between December 2019 and August 2020.

Introduction

4. The first paragraph of the introduction section is repetitive. All audience of the journal knows about the general information on sex work and sex workers.

5. "The prevalence of FSWs varies from 59 0.2% to 2.6% in Asia, 0.4% to 4.3% in sub-Saharan Africa, and 0.2% to 7.4% in Latin America 60 [4]." The prevalence of FSWs? Do you mean the prevalence of sex work? Please check and revise.

6. Please seriously avoid stigmatized terms, such as prostitutes, Age of first prostitution, etc. "According to previous studies, approximately 40 to 42 million prostitutes worldwide; about 61 80% are women between 13 and 25 [5, 6]."

7. I found some sentences very similar to other papers on abortion in female sex workers in Iran. Please revise them and check them through the paper. When I see these similarities, I don’t trust the whole article.

a. Moradi et al: "FSWs' sexual and reproductive health needs are complex due to their vulnerability to STIs and unwanted pregnancies [10]. " Karamouzian et al: " Female sex workers' (FSWs) sexual and reproductive health needs are complicated given their vulnerability to sexually transmitted infections (STIs) and unintended pregnancies [1]."

b. Moradi et al: " The health risks associated with pregnancy outcomes are ignored in FSWs who are often unrecognized as mothers [11]." Karamouzian et al: " Not often recognized as mothers, health risks associated with pregnancy outcomes in FSWs remain overlooked [2, 5]."

c. Moradi et al: " FSWs are more at risk for unwanted pregnancies and abortions than women of reproductive age in the general population [12, 13]." Khezri et al: " Female sex workers (FSWs) are at higher risk of unintended pregnancy and induced abortion, compared with women of reproductive age in the general population [1]."

d. Moradi et al: " Abortion is strictly forbidden in Iran unless there is a life-threatening medical symptom in the mother or severe fetal malformation [15, 17-19]." Khezri et al: " Induced abortion in Iran is strictly prohibited, unless there is a life-threatening medical indication in the mother or a severe foetal abnormality."

e. Moradi et al: " However, induction abortion is not uncommon. For example, in 2012, the annual induced abortion rate was estimated to be approximately 8.9 per 1000 women in the general population [20, 21]." Khezri et al: " Induced abortion is not uncommon, however; for example, in 2012 the annual induced abortion rate was estimated at approximately 8.9 per 1000 women in the general population [12,15]."

f. Moradi et al: " A 2010 study of FSWs in Iran estimated the annual abortion rate at 81 20.7 per 1000 women." Khezri et al: " A study carried out in 2010 among FSWs in Iran estimated the annual rate of abortion at 20.7 per 1000 women [16]."

h. Moradi et al: " Evidence indicates that many abortions in this population are unsafe [14]." Karamouzian et al: " Evidence suggests that many abortions among this population are unsafe,"

The introduction should be revised thoroughly. To improve the introduction, you need to provide more info and context about sex work, condom use, drug use, HIV, and abortion regarding sex workers in Iran. There are systematic reviews on these topics in FSWs in Iran that you can use and provide more context about sex workers in Iran to the journal's audience.

Method

8. "IBBS-III (integrated bio-behavioral surveillance-III)" and RDS (Respondent-Drive Sampling) should be integrated bio-behavioral surveillance-III (IBBS-III) and Respondent-Drive Sampling (RDS).

9. "Finally, 1515 FSWs participated in the study (more details are 101 in the press in the article "Behavioral and serological survey of HIV/AIDS prevalence among 102 female sex workers in Iran: A national study using respondent-driven sampling-2020")." You should cite the paper as in press paper.

10. Why did you obtain the written informed consent? The verbal informed consent sufficed to improve participant confidentiality on this illegal ground.

11. The authors correctly used an unweighted logistic regression model for analyzing RDS data. But if you use unweighted logistic regression, you should report unweighted percentages. Please report both weighted and unweighted percentages in the tables.

12. I have several concerns about the statistical analysis. First, the authors did not report the cut-off and approach for entering covariates from the bivariable regression model into the multivariable regression model. Second, entering some variables into the model is problematic. For example, the type of sexual contact with clients (vaginal and anal/oral). It is evident that only vaginal sex can lead to pregnancy, not anal or oral sex. Second, I have a hard time making conceptual relevance regarding ever alcohol use and abortion. Ever alcohol use is not even a risk factor for any condition. Even if it is significant in the analysis, I think making conceptual sense is a more critical factor to consider. Third, some variables, such as frequent use of condoms, education level of the sexual partner, marital status, and education level, are not categorized appropriately. Two or three categories are enough and make more sense. Fourth, the timeframe for some variables is unknown. Fifth, "attending team houses and hangouts to have sex with clients or finding clients (yes, no)," and "main way of client acquisition (team houses/hangouts, referrers (owners and pimps), cyberspace (via mobile, internet, and social networks) and others (parties, shopping centers, streets, parks, introduction through friends, hotels, inns and public transportation)," are overlapping variables. Lastly, be consistent about using each term. For example, ever had violent sex and experience of sexual violence. Use one term throughout.

Discussion

13. Overall, the discussion needs more work, and there should be clear policy implications different than previous studies. It would be essential to include specific and new recommendations that address the risk factor of abortion.

14. Author claim that "in a previous study, the prevalence of intentional abortion among Iranian FSWs was reported to be 35.3%, so intentional abortion among FSWs in Iran has increased by approximately 2 % between 2010 and 2020 [20, 21]." However, as the author cited, there is a paper on abortion among FSWs in Iran in 2015, which reported 46.5% of FSWs in Iran reported having had at least one induced abortion in their lifetime. The authors should take all evidence into account and then compare and discuss the current study's findings.

15. Again, similar sentences. This is not a scientific approach.

a. Moradi et al: " the prevalence of intentional abortion was reported from 11.7% in Swaziland to more than 80% in Cote d'Ivoire [22-25]." Karamouzian et al: " pregnancy termination prevalence ranging from 11.7% in Swaziland to over 80% in Cote d'Ivoire [14, 20],".

b. Moradi et al: " In addition, this prevalence was varied 202 from 21.4% to 40.0% in Cambodia [26, 27], up to 51% in China (18) and 53% in Colombia [28]." Khezri et al: " For example, the prevalence of induced abortion ranged from 21.4% to 40.0% in Cambodia [3,4], to 51% in China [14] and 53% in Colombia [22]."

c. Moradi et al: " Brothels are places where sexual services are provided according to commercial and organized rules and are controlled by a pimp. The traditional practice of brothels is based on men's demand for sexual favors and women's supply of sexual services and is illegal in Iran. Thus, FSWs in brothels may face more minor contraceptives, high-risk sexual acts such as violent sex, and engagement in high-risk environments, often leading to high-risk behaviors [34, 35]. In this study, work in a brothel and the experience of sexual violence were significantly associated with a higher probability of intentional abortion among FSWs. This finding has also been reported in several other studies that abortion is more common among non-street sex workers, such as women working in clubs, hotels, and brothels [13, 36]. The association between abortion and brothel work may reflect brothel FSWs' relatively high status and income in paying for abortion procedures or medication compared to those working on the street [13, 36]."

Khezri et al: "In Iran, operating a brothel and pimping are illegal activities; the term ‘brothel’ refers to an underground house controlled by a pimp, where sex work takes place [13]. Therefore, FSWs in brothels may face more challenges in retaining agency over their reproductive practices. Indeed, we found that working in a brothel and experiencing sexual violence were significantly associated with a higher likelihood of induced abortion among FSWs. This has also been found in many studies in different international settings, where induced abortion was more common among FSWs working in venues such as clubs or hotels [2–4,25]. A Russian study indicated that the association between induced abortion and working in a brothel might reflect a higher potential to pay for an abortion, given the higher status and income among brothel workers [2]. Moreover, FSWs in brothels may have diminished access to contraception and undergo external pressure to have an abortion, because visible pregnancy would likely affect their earning potential [26].

Limitation

16. In the limitation section, the authors only reported one limitation for the current study, and instead, they noted several strengths of their research that it is not correct.

a. "However, it should be noted that these results can still be very important and significant due to the lack of reliable information in the country. However, in addition to the limitations, the present study also has many strengths. One of the most important is country data in the behavioral and serological survey, which has tried to collect data with high reliability." There are two previous studies, and there is no lack of data on this issue. Please turn down the tone.

b. "On the other hand, sampling in this study has been done for the first time in the country using the RDS method, which is one of the essential and good sampling methods in hidden populations." This is not the first study that used RDS sampling in hidden populations in Iran, even in this population.

Conclusion

In both the Discussion AND conclusion section, the authors stated, "Based on this result, the prevalence of intentional abortion in FSWs is about seven times more than that of the general female population aged 15-50 years…". It is a wrong comparison as they compare their national cross-sectional sample with a very small-scale case-control study among women in Tehran.

6. PLOS authors have the option to publish the peer review history of their article (what does this mean?). If published, this will include your full peer review and any attached files.

Reviewer #1: No

Reviewer #2: No

---

## [Author Response · Author response to Decision Letter 0]

27 May 2022

Author's response to decision letter for (EJMR-D-21-01338): “Intentional abortion and its associated factors among female sex workers in Iran: Results from National Bio-Behavioral Surveillance-2020.”

May 23, 2022

Dear Editor,

We appreciate you and the reviewers of the “PloS One” journal for your precious time reviewing our paper and providing valuable comments. Your helpful and insightful comments led to possible improvements in the current version. The authors have carefully considered the statements and tried their best to address every one of them. We hope the manuscript, after careful revisions, meets your high standards. The authors welcome further constructive comments, if any. 

Below we provide the point-by-point responses. All modifications in the manuscript have been highlighted in yellow. 

Best Regards,

Yousef Moradi, PhD

Yousef.Moradi@muk.ac.ir

Social Determinants of Health Research Center, Research Institute for Health Development, Kurdistan University of Medical Sciences, Sanandaj 6617713446, Iran.

Reviewer #1: I would like to acknowledge the tenacity and drive required to undertake this type of research. This manuscript tackles an important Public Health issue worldwide. However, the manuscript also presents significant deficiencies, some of which I have detailed below:

The English usage in the manuscript should be thoroughly revised.

Thanks for your comment; we revised all English language of the manuscript and highlighted them in the revised manuscript. 

Abstract 

Suggesting to point out the “Background” instead of “Purpose” and add 1 or 2 lines of the background of the study. 

Thanks, done.

Page 2, line 30, “Participants,” is started with the capital word.

Thanks, done.

Page 2, line 38 Mention “type of sexual activity” instead of “type of sex.”

Thanks, done.

Introduction:

The introduction part needs to be improved with proper linking of background.

Thanks, all sections of the introduction were reviewed and added new information and highlighted in the revised manuscript.

The Social impact of abortion in FSWs in Iran should be explained.

Thanks for your comment; we added new information in the introduction section related to the social impact of abortion in FSWs and highlighted it in the revised manuscript. 

The status of intentional abortion in FSWs in some neighboring countries should be mentioned.

Thanks for your comment; we added new information in the introduction section related to other countries and highlighted it in the revised manuscript. 

Is there any current surveillance of FSWs available in Iran? If yes, please mention it with the function.

Thanks, yes, several bio-behavioral surveillance studies have been conducted in high-risk groups in Iran. These studies are performed in high-risk groups in the country every 3 to 4 times to follow high-risk behaviors. In the case of female sex workers, bio-behavioral surveillance studies have been conducted twice so far. This study is the third bio-behavioral surveillance study on female sex workers in the country.

Is there any current intervention or Government policy providing a better life to FSWs in Iran? Please briefly describe it.

Thank you. Yes, there are several service centers for sex workers in Iran. In addition to these centers, several non-governmental and governmental organizations, such as Welfare, try to provide services to these groups. Services provided to these groups include condom distribution, care and harm reduction services, and educational services.

Page 3, line 55 What type of services did you mention? 

Thanks, this section explains in the revised manuscript.

Page 3, line 61, add “age” 

Thanks, done.

Page 4, line 76 mentions the range of reproductive age 

Thanks, done.

Methods:

What’s the cause of selecting the RDS method in sampling? What’s the difference between it among the other sampling procedures? 

Thanks, RDS sampling is the best method to better and proper access to the hidden population in epidemiology studies. RDS relies on multiple waves of peer-to-peer recruitment and statistical adjustments to try and approximate random sampling. RDS only works in populations that are connected. Furthermore, the population must be large enough to sustain long referral chains without repeated participants. In our study, there were these sections. Also, the RDS method can use statistical techniques to reduce biases in the data, such as differential recruitment and differential social network sizes(1).

In which considerations you select eight cities? What’s the proportion of the FSWs in these cities?

How do you check the collinearity? Explain it in the Data analysis part

To select the provinces and cities studied in this study, most of the provinces had been studied in previous periods of bio-behavioral surveillance studies, and the results of previous studies have shown that the population of female sex workers in these cities was higher. In addition, one or two cities that have recently been shown to have sex workers as a health problem, such as Bandar Abbas, were added to these cities, bringing the total number of sampling cities to eight.

Page 7, line 133 mention the software name

Thanks, done.

Page 7, line 149 P-Value “≤ 0.05” 

Thanks, done.

Page 7, line 152 Please mention which software was used for which analysis separately.

Thanks, done.

Results: 

Page 8, line 160 add “%” after values.

Thanks, done.

Page 11, line 181 Please elaborate on the “AOR” 

Thanks, done.

Page 12, line 183 Mention “type of sexual activity” instead of “type of sex” 

Thanks, done.

Discussion:

Page 14, line 195 Replace the line “This result was indicated the …….. professionals in this area” after the “In a previous study…… 2 % between 2010 and 2020” 

Thanks, done.

Page 14, line 198 What’re the possible causes of to increase in the prevalence in Iran? Mention it.

Thanks for your comment; we added related information and studies result in this section and highlighted them in the revised manuscript. 

Conclusion:

Mention some significant risk factors of the current study.

Thanks for your comment; we added some significant risk factors in the abstract and article conclusion and highlighted them in the revised manuscript.

References

Please check the reference style of the journal and maintain the format in all references.

Thanks, done.

Example: 1. Raifman S, DeVost MA, Digitale JC, Chen Y-H, Morris MD. Respondent-Driven Sampling: a Sampling Method for Hard-to-Reach Populations and Beyond. Current Epidemiology Reports. 2022:1-10.

Table

Mention the elaboration of the shorts forms below the tables that were used in tables.

Thanks, done.

Reviewer #2: This topic is interesting due to limited attention to the sexual and reproductive health needs of female sex workers worldwide. However, I have several concerns to be addressed to refine the manuscript. My main concern is that this replication of the analysis of the previous surveys on the history of induced abortion among FSW is not a significant contribution to the literature and does not add much to what we already know. I’d consider adding to the depth of the analysis and also evaluating contraceptive use practices of FSW in Iran. Please see my comments in the following.

1. Overall, the paper is poorly written. I would suggest requesting a style and grammar review before submitting it to the journal.

Thanks, done.

Abstract

2. Authors stated that "In addition to estimating the prevalence of intentional abortion in Iranian Female sex workers (FSWs), this study identified related factors using the data of a national study." As far as I know, two previous studies on female sex workers in Iran examined abortion and its associated factors. What is the added value of this paper to our understanding of abortion in this population in Iran? 

So far, two bio-behavioral surveillance studies have been conducted on sexually active women, which is the third study in this group. One of the previous two studies on abortion in FSWs also published good information. The present study was conducted and published by health policymakers and researchers to complete the published data and to compare and review the abortion process in these groups. In addition, the present study results show the nature of bio-behavioral surveillance studies in the country, which leads to the investigation of the consequences of the study, such as abortion in high-risk groups, especially FSWs.

3. "….in December 2019 and August 2020….". It should be between December 2019 and August 2020.

Thanks, done.

Introduction

4. The first paragraph of the introduction section is repetitive. All audience of the journal knows about the general information on sex work and sex workers.

Thank you for your comment. The First paragraph was deleted. 

5. "The prevalence of FSWs varies from 0.2% to 2.6% in Asia, 0.4% to 4.3% in sub-Saharan Africa, and 0.2% to 7.4% in Latin America [4]." The prevalence of FSWs? Do you mean the prevalence of sex work? Please check and revise.

Thank you for your comment. We checked the references, and this prevalence is confirmed in various countries. Also, we added more references in the revised manuscript related to this sentence. 

6. Please seriously avoid stigmatized terms, such as prostitutes, Age of first prostitution, etc. "According to previous studies, approximately 40 to 42 million prostitutes worldwide; about 61 80% are women between 13 and 25 [5, 6]."

Thank you for your comment. The First paragraph was edited and highlighted in the revised manuscript. 

7. I found some sentences very similar to other papers on abortion in female sex workers in Iran. Please revise them and check them through the paper. When I see these similarities, I don’t trust the whole article.

Thank you for your comment. All sections were rechecked and highlighted all changes in the revised manuscript. 

a. Moradi et al.: "FSWs' sexual and reproductive health needs are complex due to their vulnerability to STIs and unwanted pregnancies [10]. " Karamouzian et al.: " Female sex workers (FSWs) sexual and reproductive health needs are complicated given their vulnerability to sexually transmitted infections (STIs) and unintended pregnancies [1]."

Thank you for your comment. This sentence was checked and changed in the revised manuscript. 

b. Moradi et al.: " The health risks associated with pregnancy outcomes are ignored in FSWs who are often unrecognized as mothers [11]." Karamouzian et al.: " Not often recognized as mothers, health risks associated with pregnancy outcomes in FSWs remain overlooked [2, 5]."

Thank you for your comment. This sentence was checked and changed in the revised manuscript. 

c. Moradi et al.: " FSWs are more at risk for unwanted pregnancies and abortions than women of reproductive age in the general population [12, 13]." Khezri et al.: " Female sex workers (FSWs) are at higher risk of unintended pregnancy and induced abortion, compared with women of reproductive age in the general population [1]."

Thank you for your comment. This sentence was checked and changed in the revised manuscript. 

d. Moradi et al.: " Abortion is strictly forbidden in Iran unless there is a life-threatening medical symptom in the mother or severe fetal malformation [15, 17-19]." Khezri et al.: " Induced abortion in Iran is strictly prohibited unless there is a life-threatening medical indication in the mother or a severe fetal abnormality."

Thank you for your comment. This sentence was checked and changed in the revised manuscript. 

e. Moradi et al.: " However, induction abortion is not uncommon. For example, in 2012, the annual induced abortion rate was estimated to be approximately 8.9 per 1000 women in the general population [20, 21]." Khezri et al.: " Induced abortion is not uncommon, however; for example, in 2012, the annual induced abortion rate was estimated at approximately 8.9 per 1000 women in the general population [12,15]."

Thank you for your comment. This sentence was checked and changed in the revised manuscript. 

f. Moradi et al.: " A 2010 study of FSWs in Iran estimated the annual abortion rate at 81 20.7 per 1000 women." Khezri et al.: " A study carried out in 2010 among FSWs in Iran estimated the annual rate of abortion at 20.7 per 1000 women [16]."

Thank you for your comment. This sentence was checked and changed in the revised manuscript. 

h. Moradi et al.: " Evidence indicates that many abortions in this population are unsafe [14]." Karamouzian et al: " Evidence suggests that many abortions among this population are unsafe,"

Thank you for your comment. This sentence was checked and changed in the revised manuscript. 

The introduction should be revised thoroughly. To improve the introduction, you need to provide more info and context about sex work, condom use, drug use, HIV, and abortion regarding sex workers in Iran. There are systematic reviews on these topics in FSWs in Iran that you can use and provide more context about sex workers in Iran to the journal's audience.

Thanks for your attention; we edited all sections of the introduction and added new information and references in the revised manuscript. 

Method

8. "IBBS-III (integrated bio-behavioral surveillance-III)" and RDS (Respondent-Drive Sampling) should be integrated bio-behavioral surveillance-III (IBBS-III) and Respondent-Drive Sampling (RDS).

Thanks, done.

9. "Finally, 1515 FSWs participated in the study (more details are 101 in the press in the article "Behavioral and serological survey of HIV/AIDS prevalence among 102 female sex workers in Iran: A national study using respondent-driven sampling-2020")." You should cite the paper as in press paper.

Thanks, done.

10. Why did you obtain the written informed consent? The verbal informed consent sufficed to improve participant confidentiality on this illegal ground.

The World Health Organization funded the present study. All studies conducted in high-risk groups, especially sexual minorities and FSWs funded by the organization, must complete the WHO consent form in person and in writing. In this study, according to the instructions of the WHO, the informed consent form of this organization was read to all FSWs, and the written consent of these people to participate in the study was received.

11. The authors correctly used an unweighted logistic regression model for analyzing RDS data. But if you use unweighted logistic regression, you should report unweighted percentages. Please report both weighted and unweighted rates in the tables.

Thanks, after collecting the samples in the RDS method, analysis consistent with RDS should be used. The weight percentages must be reported through the special RDS software. For multivariate analysis, weight analysis is not recommended according to the latest studies. We did not report non-weight percentages for three other reasons. First, these percentages are less valid than weight percentages. Second, the tables should not confuse readers, and third, non-weight rates can be calculated through the reported frequencies. To clarify this ambiguity, we added a description of the method. But if it is recommended to mention the non-weight percentages again with this explanation, please let us know so that we can rewrite the tables.

12. I have several concerns about the statistical analysis. First, the authors did not report the cut-off and approach for entering covariates from the bivariable regression model into the multivariable regression model. Second, entering some variables into the model is problematic. For example, the type of sexual contact with clients (vaginal and anal/oral). Only vaginal sex can lead to pregnancy, not anal or oral sex. Second, I have a hard time making conceptual relevance regarding alcohol use and abortion. Ever alcohol use is not even a risk factor for any condition. Even if it is significant in the analysis, I think making conceptual sense is a more critical factor to consider. Third, some variables, such as frequent use of condoms, education level of the sexual partner, marital status, and education level, are not categorized appropriately. Two or three categories are enough and make more sense. Fourth, the timeframe for some variables is unknown. Fifth, "attending team houses and hangouts to have sex with clients or finding clients (yes, no)," and "main way of client acquisition (team houses/hangouts, referrers (owners and pimps), cyberspace (via mobile, internet, and social networks) and others (parties, shopping centers, streets, parks, introduction through friends, hotels, inns and public transportation)," are overlapping variables. Lastly, be consistent about using each term. For example, ever had violent sex and experienced sexual violence. Use one term throughout.

Thanks for the valuable comments. We edited the article based on your comments one by one as follows:

1. We explained the selection of variables in the multivariate model to the method. 

2. Indeed, people with a history of oral or anal sex have also had vaginal sex. There are several hypotheses about the relationship between abortion and the type of sex. Do Iranian women tend to have other types of sex to avoid pregnancy? Is there a history of different kinds of sex for more pleasure, a tendency to experience a variety of behaviors in riskier people, and not paying attention to contraception? Based on your comment in the discussion, we mentioned this hypothesis to explain the conceptual relevance between the variables to the reader.

3. Alcohol consumption also has special conditions because it is banned in Iran. Do people with a history of alcohol use have specific characteristics that can be riskier? These hypotheses and the conditions of Iran can explain the conceptual relevance between the variables to the reader. We mentioned this hypothesis based on your comment in the discussion.

4. The main way of client acquisition seems to overlap: attending team houses and hangouts to have sex with clients or finding clients. But the first variable considers the history of attending team houses and spots, and the second variable refers to the main way of client acquisition. We changed the titles of the variables to clear up this ambiguity for the readers. 

5. We equated the terms ever had violent sex and experience of sexual violence. 

Discussion

13. Overall, the discussion needs more work, and there should be clear policy implications different than previous studies. It would be essential to include specific and new recommendations that address the risk factor of abortion.

Thanks for your comments. We edited the discussion section based on your comments and highlighted it in the revised manuscript.

14. Author claim that "in a previous study, the prevalence of intentional abortion among Iranian FSWs was reported to be 35.3%, so intentional abortion among FSWs in Iran has increased by approximately 2 % between 2010 and 2020 [20, 21]." However, as the author cited, there is a paper on abortion among FSWs in Iran in 2015, which reported 46.5% of FSWs in Iran reported having had at least one induced abortion in their lifetime. The authors should take all evidence into account and then compare and discuss the current study's findings.

Thanks to the honorable referee. In the discussion section of this article, the authors tried to include all the articles worked on in Iran on the subject. The article of Khezri et al. In Iran reported a raw prevalence of abortion in the population of FSWs equal to 46.5%. Still, the present study’s prevalence was equal to 37.3%. The difference between the present study and the study of Khezri et al. is that in the present study, the prevalence of weight gain was used to report the prevalence of abortion in the population of FSWs in Iran. In contrast, in the study of Khezri et al., this outcome was reported by the raw prevalence. Due to the lack of indicators affecting the prevalence in this study, the prevalence is higher than in the present study.

To apply the opinions of the esteemed referee in the discussion section, the following sentence was added and highlighted.

In the study of Khezri et al. conducted in 13 provinces of Iran, the prevalence of abortion in Iranian FSWs was 46.5%, while the prevalence in the present study was 37.3%. One of the essential reasons for this difference in the results of these two studies, both of which have been conducted in Iran at different times, is the difference in the selected provinces and how to report or calculate the prevalence of abortion. In the study by Khezri et al., 13 areas were selected, while in the present study, eight provinces were surveyed. The provinces that may have the highest prevalence of female FSWs were studied in a study by Khezri et al. On the other hand, the prevalence of weight was used to report abortion in the present study. In contrast, the study of Khezri et al. used the raw prevalence to report abortion.

15. Again, similar sentences. This is not a scientific approach.

a. Moradi et al.: " the prevalence of intentional abortion was reported from 11.7% in Swaziland to more than 80% in Cote d'Ivoire [22-25]." Karamouzian et al: " pregnancy termination prevalence ranging from 11.7% in Swaziland to over 80% in Cote d'Ivoire [14, 20],".

Thanks for your attention; we edited these sections in the discussion and highlighted them in the revised manuscript.

b. Moradi et al: " In addition, this prevalence was varied 202 from 21.4% to 40.0% in Cambodia [26, 27], up to 51% in China (18) and 53% in Colombia [28]." Khezri et al: " For example, the prevalence of induced abortion ranged from 21.4% to 40.0% in Cambodia [3,4], to 51% in China [14] and 53% in Colombia [22]."

Thanks to the esteemed referee, the sentences mentioned below were changed and highlighted in the discussion section.

According to the results of previous studies, the prevalence of abortion in the population of FSWs varies in different parts of the world. In African countries, this prevalence is up to 80%, in European countries, up to 11.7%, and in Asian countries, up to 51%.

c. Moradi et al.: " Brothels are places where sexual services are provided according to commercial and organized rules and are controlled by a pimp. The traditional practice of brothels is based on men's demand for sexual favors and women's supply of sexual services and is illegal in Iran. Thus, FSWs in brothels may face more minor contraceptives, high-risk sexual acts such as violent sex, and engagement in high-risk environments, often leading to high-risk behaviors [34, 35]. In this study, work in a brothel and the experience of sexual violence were significantly associated with a higher probability of intentional abortion among FSWs. This finding has also been reported in several other studies that abortion is more common among non-street sex workers, such as women working in clubs, hotels, and brothels [13, 36]. The association between abortion and brothel work may reflect brothel FSWs' relatively high status and income in paying for abortion procedures or medication compared to those working on the street [13, 36]."

Khezri et al.: "In Iran, operating a brothel and pimping are illegal activities; the term ‘brothel’ refers to an underground house controlled by a pimp, where sex work takes place [13]. Therefore, FSWs in brothels may face more challenges in retaining agency over their reproductive practices. Indeed, we found that working in a brothel and experiencing sexual violence were significantly associated with a higher likelihood of induced abortion among FSWs. This has also been found in many studies in different international settings, where induced abortion was more common among FSWs working in clubs or hotels [2–4,25]. A Russian study indicated that the association between induced abortion and working in a brothel might reflect a higher potential to pay for an abortion, given the higher status and income among brothel workers [2]. Moreover, FSWs in brothels may have diminished access to contraception and undergo external pressure to have an abortion because visible pregnancy would likely affect their earning potential [26].

Thanks for your attention; we edited these sections in the discussion and highlighted them in the revised manuscript.

Limitation

16. In the limitation section, the authors only reported one limitation for the current study, and instead, they noted several strengths of their research that it is not correct.

a. "However, it should be noted that these results can still be very important and significant due to the lack of reliable information in the country. However, in addition to the limitations, the present study also has many strengths. One of the most important is country data in the behavioral and serological survey, which has tried to collect data with high reliability." There are two previous studies, and there is no lack of data on this issue. Please turn down the tone.

Thanks for your attention; we edited this section and highlighted it in the revised manuscript.

b. "On the other hand, sampling in this study has been done for the first time in the country using the RDS method, one of the essential and good sampling methods in hidden populations." This is not the first study that used RDS sampling in hidden populations in Iran, even in this population.

Thanks for your attention; we edited this section and highlighted it in the revised manuscript.

Conclusion

In the Discussion AND conclusion section, the authors stated, "Based on this result, the prevalence of intentional abortion in FSWs is about seven times more than that of the general female population aged 15-50 years…". It is a wrong comparison as they compare their national cross-sectional sample with a very small-scale case-control study among women in Tehran.

Thanks for your attention; we edited this section as follows and highlighted it in the revised manuscript.

Based on this result, the prevalence of intentional abortion in FSWs is higher than in the general female population aged 15-50 years, which indicates a warning issue in the country's public health and needs the great attention of policymakers.

---

## [Decision Letter · Decision Letter 1]

15 Jun 2022

PONE-D-22-07026R1Intentional abortion and its associated factors among female sex workers in Iran: results from national bio-behavioral surveillance-2020PLOS ONE

Dear Dr. Moradi,

Thank you for submitting your manuscript to PLOS ONE. After careful consideration, we feel that it has merit but does not fully meet PLOS ONE’s publication criteria as it currently stands. Therefore, we invite you to submit a revised version of the manuscript that addresses the points raised during the review process.

We look forward to receiving your revised manuscript.

Kind regards,

Hamid Sharifi

Academic Editor

PLOS ONE

Additional Editor Comments:

Dear Dr. Moradi,

Thanks so much for submitting the revised manuscript. Unfortunately, the reviewers put more comments and they believed your responses and modifications were not satisfactory in the grammatical and scientific contexts.

Please modify the file based on the comments of the previous and current versions.

Best Regards

Hamid Sharifi

Reviewers' comments:

Reviewer's Responses to Questions

**Comments to the Author**

1. If the authors have adequately addressed your comments raised in a previous round of review and you feel that this manuscript is now acceptable for publication, you may indicate that here to bypass the “Comments to the Author” section, enter your conflict of interest statement in the “Confidential to Editor” section, and submit your "Accept" recommendation.

Reviewer #1: All comments have been addressed

Reviewer #2: (No Response)

2. Is the manuscript technically sound, and do the data support the conclusions?

Reviewer #1: Yes

Reviewer #2: No

3. Has the statistical analysis been performed appropriately and rigorously? 

Reviewer #1: Yes

Reviewer #2: No

4. Have the authors made all data underlying the findings in their manuscript fully available?

Reviewer #1: Yes

Reviewer #2: (No Response)

5. Is the manuscript presented in an intelligible fashion and written in standard English?

Reviewer #1: Yes

Reviewer #2: (No Response)

6. Review Comments to the Author

Reviewer #1: Abstract

Page 2 line 31 Change “Participants” to’ ‘participants’

Page 2 line 40 Mention “type of sexual activity” instead of “type of sex”

Introduction:

Page 4 line 73 add “age” after ‘between’

Page 4 line 88 mention the range of reproductive age

Methods:

Page 8 line 179 mention the software name

Page 9 line 198 P-Value “≤ 0.05”

Page 10 line 204-205 Please mention which software was used for which analysis separately.

Results:

Page 10 line 213 add “%” after values.

Page 14 line 234 Please elaborate the “AOR”

Page 14 line 236 Mention “type of sexual activity” instead of “type of sex”

Reviewer #2: The authors have not addressed previous comments and concerns appropriately. They did not revise their statistical analysis and problematic variables. The English usage in the manuscript has not been edited carefully. While the authors did not revise some of the similar sentences, they mentioned that we edited them. The discussion section also has several problematic statements. Overall, the authors did not try their best to improve the manuscript and even undermined it.

7. PLOS authors have the option to publish the peer review history of their article (what does this mean?). If published, this will include your full peer review and any attached files.

Reviewer #1: **Yes: **Pronesh Dutta

Reviewer #2: No

---

## [Author Response · Author response to Decision Letter 1]

30 Jul 2022

Author's response to decision letter for (EJMR-D-21-01338): “Intentional abortion and its associated factors among female sex workers in Iran: Results from National Bio-Behavioral Surveillance-2020.”

July 30, 2022

Dear Editor,

We appreciate you and the reviewers of the “PloS One” journal for your precious time reviewing our paper and providing valuable comments. Your helpful and insightful comments led to possible improvements in the current version. The authors have carefully considered the statements and tried their best to address every one of them. We hope the manuscript, after careful revisions, meets your high standards. The authors welcome further constructive comments if any. 

Below we provide the point-by-point responses highlighted in red. 

Best Regards,

Yousef Moradi, PhD

Yousefmoradi211@yahoo.com

Social Determinants of Health Research Center, Research Institute for Health Development, Kurdistan University of Medical Sciences, Sanandaj 6617713446, Iran.

Review Comments to the Author

We've checked your submission and before we can proceed, we need you to address the following issues:

1. Thank you for updating your Data Availability statement to:

"The data supporting this study's findings cannot be shared publicly because of ethical restrictions involving patient information. Data are available on request from the Social Determinants of Health Research Center, Research Institute for Health Development, Kurdistan University of Medical Sciences after appropriate protocol submission to the institution’s office of Human Research Ethics Committee. (contact via sdhkurdistan@gmail.com) for researchers who meet the criteria for access to confidential data."

Can you please clarify who is the recipient of the provided contact email "sdhkurdistan@gmail.com" and what their role is at the Social Determinants of Health Research Center, Research Institute for Health Development, Kurdistan University of Medical Sciences?

Response:

Thank you so much. The data supporting this study's findings cannot be shared publicly because of ethical restrictions involving patient information. Data are available on request from the Social Determinants of Health Research Center, Research Institute for Health Development, Kurdistan University of Medical Sciences after appropriate protocol submission to the institution’s office of Human Research Ethics Committee. (Contact via sdhkurdistan@gmail.com ) for researchers who meet the criteria for access to confidential data. 

The Social Determinant of Health Research Center began its work in 2012. During two years this center published over 100 articles in international and national journals, and it has carried out many research projects in the public health issues. Often, these projects were funded by WHO, UNAIDS, UNDP, and Iranian Ministry of Health. This center has a Research Council Members for deciding about projects and related their data. This Research Council Members included several experts and researchers related to public health issues (https://muk.ac.ir/Page?pageId=8714). The Dr. Bakhtiar Piroozi is head of center and he responsible to receive and response all requests.

---

## [Editor Report · Decision Letter 2]

15 Aug 2022

Intentional abortion and its associated factors among female sex workers in Iran: Results from National Bio-Behavioral Surveillance-2020

PONE-D-22-07026R2

Dear Dr. Moradi

We’re pleased to inform you that your manuscript has been judged scientifically suitable for publication and will be formally accepted for publication once it meets all outstanding technical requirements.

Kind regards,

Hamid Sharifi

Academic Editor

PLOS ONE
---

## [Editor Report · Acceptance letter]

18 Aug 2022

PONE-D-22-07026R2 

Intentional abortion and its associated factors among female sex workers in Iran: Results from National Bio-Behavioral Surveillance-2020 

Dear Dr. Moradi:

I'm pleased to inform you that your manuscript has been deemed suitable for publication in PLOS ONE. Congratulations! Your manuscript is now with our production department. 

Kind regards, 

on behalf of

Dr. Hamid Sharifi 

Academic Editor

PLOS ONE